# Implementation of Objective Structured Clinical Examination on Diagnostic Musculoskeletal Ultrasonography Training in Undergraduate Traditional Korean Medicine Education: An Action Research

**DOI:** 10.3390/diagnostics12071707

**Published:** 2022-07-13

**Authors:** Eunbyul Cho, Young-Min Han, Yeonseok Kang, Jae-Hyo Kim, Min-Seop Shin, Myungjin Oh, Nam-Geun Cho, Hyun-Jong Jung, Jungtae Leem

**Affiliations:** 1College of Korean Medicine, Wonkwang University, 460, Iksan-daero, Iksan 54538, Korea; chostar427@gmail.com (E.C.); yeonkang@wku.ac.kr (Y.K.); medicdog@wku.ac.kr (J.-H.K.); choandle@hanmail.net (N.-G.C.); 2Cheoncheon Public Health Center Branch, 35, Songtan-ro, Cheoncheon-myeon, Jangsu 55607, Korea; nontroppo21@hanmail.net; 3Center for International Cooperation & Education on Korean Medicine, Wonkwang University, Iksan 54538, Korea; 4Traditional Medicine Research Center, Wonkwang University, 460, Iksan-daero, Iksan 54538, Korea; 5Shin Min Seop Korean Medicine Clinic, 163, Geomapyeong-ro, Jeonju 55056, Korea; kahn815@hanmail.net; 6Keumkang Korean Medical Clinic, 714, Naesu-ro, Cheongju 28145, Korea; kkomc@hanmail.net

**Keywords:** carpal tunnel syndrome, diagnostic ultrasound, learning outcomes, OSCE, survey

## Abstract

This study aimed to report the specific methods and investigate the educational effects of diagnostic musculoskeletal ultrasound training and the Objective Structured Clinical Examination (OSCE) for traditional medicine students. Scanning volar wrist and diagnosing carpal tunnel syndrome were selected for musculoskeletal ultrasound to train students to use the basic functions of the ultrasound device and scan various structures including tendons, nerves, and arteries. The students were divided into two groups: one group had 8 weeks of training with mock OSCE experience and received feedback about their scan images, and the other group had 3 weeks of training with flipped learning. The OSCE was implemented on the last day of the training. The subjective learning outcomes were analyzed as students’ evaluation with a 5-point scale, and the objective learning outcomes were analyzed using OSCE scores evaluated with a pre-validated checklist. Of the 111 students, 60 (54.1%) responded to the questionnaire. Overall satisfaction with this ultrasound training was high (4.5 ± 0.60). The average OSCE score in the 8-week group was significantly higher than that in the 3-week group. The students’ self-assessment showed no significant differences between the two groups. Proficiency in using ultrasound is affected by the practice time and feedback. Ultrasound training should be further expanded as a required curriculum to meet students’ needs and achieve learning objectives in the clinical skills education of Korean medicine colleges. Further studies are needed on ultrasound education, especially guided interventions for traditional medicine students.

## 1. Introduction

Modern medical education has undergone a change from apprenticeship to lecture-oriented education and to systematic education combining theory and practice [1]. In recent years, outcome-based education has been emphasized, focusing on the competencies that students need to achieve, including their actual performance [2]. In this context, the limitations and issues of validity, reliability, and diversity of the traditional clinical examination handling a single long case at the bedside have been raised. To overcome these limitations, the Objective Structured Clinical Examination (OSCE), a clinical competence-based assessment, was developed and has been used worldwide for healthcare education. In the OSCE, learning outcomes and tasks to be performed by the examinee are defined for each station, and the examinee’s performance is evaluated according to predetermined rating scales or checklists [3]. Since its introduction to medical education, it has shown positive effects, including feasibility, flexibility with fewer cultural and geographic limitations, assessment of various learning outcomes, and the possibility of using them for both formative and summative purposes [4]. Ultrasound education is often covered in undergraduate medical education because the use of ultrasound devices depends on the perceived skills of the user. In many cases, ultrasonography is used to teach anatomy, procedural skills, and clinical applications. The OSCE has been used to assess technical skills in ultrasound [5].

OSCE has also been adopted in traditional medicine. In traditional medicine, the introduction of OSCE is expected to enhance clinical competency and strengthen the authenticity of education [6,7,8]. Studies on ultrasound to scan the anatomical structure of acupoints for accurate diagnosis and safe treatment [9,10,11], cases utilizing ultrasound for diagnosis and treatment [12,13], and the efficacy of ultrasound-guided intervention [14,15] have been reported by traditional medicine physicians. Nevertheless, in traditional medicine education, to date, no evidence on standardized ultrasound education and OSCE on musculoskeletal diseases has been published to our knowledge. Although musculoskeletal disorders constitute a significant portion of traditional medical care [16,17], traditional medical education does not meet the educational and clinical needs of the utilization of ultrasound devices.

Meanwhile, many practical studies have been applied in education to investigate the unmet needs of students and to apply them to improve the educational field. Action research that measures and monitors the effectiveness of the educational curriculum development and application and revises education plans is a necessary process for improving educational efficiency and student satisfaction in the newly introduced education field [18,19,20].

Therefore, in this study, we implemented diagnostic musculoskeletal ultrasound training and OSCE, focusing on the median nerve and carpal tunnel, which are included in the musculoskeletal learning objectives of the previously developed longitudinal ultrasound curricula mentioned in the study by Tarique [5]. After completing both training and assessment, we conducted a questionnaire to investigate the satisfaction with the training, suggestions to improve, and unmet needs. To this end, we developed and validated an OSCE checklist and questionnaire for students in a preliminary study [21]. This study aimed to describe in detail the case of musculoskeletal ultrasound training and the OSCE for senior students at the College of Korean Medicine. In addition, we suggest the satisfaction, unmet needs, and points for future improvements of ultrasound education for students of traditional Korean medicine using a validated questionnaire.

## 2. Materials and Methods

### 2.1. Course of Diagnostic Musculoskeletal Ultrasound Focused on Volar Wrist and Carpal Tunnel Syndrome

The training was initially designed as a face-to-face lecture once a week for 8 weeks by dividing the whole participant sample into three groups. However, due to the spread of COVID-19, only one group was educated for 8 weeks. For the other two groups, the lecture period was reduced to 3 weeks. Thus, a flipped learning curriculum was implemented to compensate for the loss of lecture time. The learning objectives were set to “I can use an ultrasonic device” and “I can examine structures of the body with ultrasound and evaluate their clinical implications” [21]. Scanning the volar wrist and diagnosing carpal tunnel syndrome were selected as educational topics so that the learners could use functions such as probe selection, depth and focus adjustment, distance and cross-sectional area measurement, body-part scanning, and inference of clinical significance. The lecture content included ultrasound principles, indications, artifacts, how to use ultrasound devices, volar wrist anatomy, scan of the volar wrist, and diagnostic criteria for carpal tunnel syndrome. The structures that can be scanned from the volar wrist are as follows: (1) scaphoid, (2) pisiform, (3) median nerve, (4) ulnar artery, (5) flexor retinaculum, (6) flexor pollicis longus, (7) ulnar nerve, (8) flexor digitorum superficialis and flexor digitorum profundus, (9) flexor carpi radialis, and (10) flexor carpi ulnaris (Figure 1). The two diagnostic criteria for carpal tunnel syndrome used in ultrasound training are as follows: (1) the cutoff value of the vertical distance between the line connecting the top of the scaphoid and the pisiform in the proximal carpal tunnel and the apex of the retinaculum is between 3 and 4 mm [22,23,24] (Figure 2); (2) the cross-sectional area of the median nerve proximal to the tunnel inlet is 10 mm^2^ or more [25,26,27] (Figure 3).

The preceptor in charge of this lecture has experience in two ultrasound clinical research studies [9,11] and in performing multiple OSCEs [7,8]. The lecture was designed with the advice of two traditional medicine doctors who obtained a certification (Registered in Musculoskeletal Sonography (RMSK)) hosted by the American Registry for Diagnostic Medical Sonography (ARDMS) [28]. The 8-week face-to-face lecture consisted of a lecture class, training, and OSCE. In the training, each member of the group of 4–5 people was able to directly use the ultrasound for approximately 40 min, twice. In the first training, the preceptor performed live demonstrations in each group, and then the students had an opportunity for a hands-on experience with each other under one-on-one instruction by the preceptor. In the second training, the students performed the entire procedure with peer assessment, similar to a mock OSCE, under the supervision of a preceptor. Self-directed practice at the clinical skills center was recommended. The students were asked to submit scan prints of structures of the volar wrist with markings and two methods for diagnosing carpal tunnel syndrome. The preceptor confirmed each student’s scanned image and provided feedback.

For the 3-week group, the students were required to take online lectures in advance. Hands-on practice and OSCE were implemented for 3 weeks. The online lecture was designed to include steps 1 to 3 of George’s five-step method of teaching clinical skills [29]. The principle of ultrasound, how to use the ultrasound device, the volar wrist structure that can be scanned at the scaphoid–pisiform level, and the scanning method according to the two diagnostic criteria for carpal tunnel syndrome were explained (step 1: overview). The entire procedure was shown in a 5 min video (step 2: preceptor demonstration). The procedure was repeated with a detailed description of each procedure (step 3: preceptor’s description of each step in the process). The time to practice ultrasound in person under the supervision of the preceptor was 1 h per group of 4–5 people.

An OSCE was conducted on the last day of training for all groups. Out of a total of 111 students, 110 participated in the OSCE; one student failed to perform the OSCE due the adverse effects of the coronavirus disease 2019 (COVID-19) vaccination. The ultrasound device used for education was the SonoAce R7 system (Samsung Medison, Seoul, Korea), and the probe was a linear-array 6–12 MHz transducer (L5-13IS, Samsung Medison, Seoul, Korea). The learning objectives of education and the validated OSCE checklist and post-education questionnaire were presented in our previous study [21].

### 2.2. Study Design and Participants

We conducted a survey with 111 undergraduate students from a single college in Korean medicine. After ultrasound training and completion of the OSCE, a questionnaire was created online (SurveyMonkey, San Mateo, CA, USA) and delivered to a student representative to recruit survey participants. The survey was conducted from 30 September 2021 to 5 October 2021; 58 participants who completed the survey and agreed to collect cell phone information were given a gift card worth 4 USD as an incentive. This study was approved by the Institutional Review Board of Wonkwang University (WKIRB-202109-SB-069).

### 2.3. Statistical Analysis

Descriptive statistical analyses were performed using SPSS version 21.0 (SPSS, IBM Corp. 2019, Armonk, NY, USA). We tried to analyze the difference in questionnaire scores and self-evaluation scores after training between the 8-week group and the 3-week group who responded to the questionnaire. As a result of the normality test, both the Kolmogorov–Smirnov test and the Shapiro–Wilk test showed a significance probability of less than 0.05, which did not satisfy normality, so the Mann–Whitney U test was performed. The Mann–Whitney U test was used to analyze whether there was a significant difference in the total OSCE scores of 39 patients in the 8-week group and 71 patients in the 3-week group who underwent OSCE. Statistical significance was set at *p* < 0.05.

## 3. Results

### 3.1. Participants

Of the 111 students who completed the curriculum, 60 responded to the questionnaire (54.1%), 37 (51.4%) practiced for 3 weeks, and 23 (59.0%) practiced for 8 weeks. Prior to this ultrasound education, 12 students had experience using an ultrasound diagnostic device and 48 students had no experience.

### 3.2. Students’ Opinions toward Ultrasound Training and OSCE

Most of the respondents replied that their understanding of the anatomical physiology of the carpal tunnel was improved through ultrasound education (4.75 ± 0.44), the ultrasound OSCE motivated independent practical training (4.45 ± 0.53), they were generally satisfied with this ultrasound training (4.5 ± 0.60), and that they had intentions of studying ultrasound more in the future (4.57 ± 0.56) (Table 1). The opinion that the 5 min time limit for the ultrasound OSCE was sufficient was more than the opinion that it was insufficient, and the opinion on whether the ultrasound OSCE was difficult was neutral. Using the Mann–Whitney U test, the 8-week group (3.74 ± 0.92) scored significantly higher than the 3-week group (2.97 ± 1.30) only for the item “Was the ultrasound OSCE time limit (5 min) enough for you?” (*p* = 0.019); there were no significant differences among the other items. The most common area for further learning in musculoskeletal ultrasound was the shoulder (29 cases), followed by the abdomen (14), knee (13), and ankle (6). There was also an opinion that they wanted to learn guide needling by selecting areas to be careful about during the acupotomy procedure.

### 3.3. Students’ Self-Assessment

To assess the subjective learning outcomes, the students who completed the curriculum were asked to self-evaluate their ability to use ultrasound and their attitude (see Table 2). As for the self-evaluation score, “Using basic functions of ultrasound” was the highest (4.52 ± 0.70), while “Being considerate of the subject during the ultrasound examination” (4.35 ± 0.66) and “Identifying the anterior structure of the wrist through ultrasound” (4.35 ± 0.71) were relatively low. Although the 8-week group’s average score was higher than that of the 3-week group for all items, there was no significant difference between the two groups using the Mann–Whitney U test (*p* > 0.05).

### 3.4. Students’ OSCE Score

Table 3 shows the students’ average scores and standard deviations for each OSCE item. Among the 15 OSCE checklist items, the achievement of “gel application” was the highest (1.00 ± 0.00), and the achievement of “hand hygiene after examination” was the lowest (0.30 ± 0.66). The OSCE total score ranged from 5 to 28, with an overall average of 19.00. The average score of the 3-week group was 16.68, and that of the 8-week group was 23.23. Kolmogorov–Smirnov (*p* = 0.002 for 8-week group) and Shapiro–Wilk (*p* = 0.010 for 8-week group) tests did not satisfy the normality test; thus, the Mann–Whitney U test was performed. As a result of the Mann–Whitney U test, the average OSCE total score of the two groups was significantly different (*p* < 0.001).

## 4. Discussion

### 4.1. Summary of Findings

This study reported the results of a survey on ultrasound training and students’ learning outcomes after training diagnostic musculoskeletal ultrasound focused on volar wrist and carpal tunnel syndrome and implementing the OSCE for senior students majoring in traditional medicine. Overall, the level of satisfaction with the education was high throughout the survey, and it was confirmed that ultrasound training may contribute not only to the technical ability to handle ultrasound devices, but also to an increased understanding of anatomy and physiology. In addition, due to the local spread of COVID-19, the learners were inevitably (non-randomized) divided into two groups: one that practiced for 8 weeks and another that practiced for 3 weeks. There was no significant difference between the two groups in the subjective self-evaluation, but the average OSCE score of the group that practiced for 8 weeks was significantly higher, and the group that practiced for 3 weeks had a low positive response to whether the OSCE time limit (five minutes) was sufficient. These results suggest that the proficiency of the 3-week group was lower than the 8-week group. These results suggest that ultrasound education is essential for outcome-based clinical skills training in Korean medicine.

### 4.2. Debates

Ultrasound is non-invasive and safe, and as technology advances, ultrasound probes have become more precise, increasing their ability to observe human structures. Recently, the use of point-of-care ultrasound, taking ultrasound to a patient’s bedside, acquiring images in real time, and using the technique in connection with the patient’s symptoms and signs have been increasing [30]. However, most of the reported ultrasound education courses at medical schools deal with internal organs, such as the cardiovascular system and abdomen, with a relatively low proportion in the musculoskeletal system [31,32]. Since musculoskeletal ultrasound enables the observation of the structure of the skin and subcutaneous tissue, muscles and tendons, nerves and blood vessels, and bones and joints, and detects lesions in real time, it is highly useful in treating patients with nervous or musculoskeletal diseases. In particular, the median nerve and carpal tunnel are important enough to be mentioned as examples of observations in musculoskeletal ultrasound in Tarique’s study, which reviewed the ultrasound curriculum in undergraduate medical education [5]. More than 20,000 people visit traditional Korean medicine clinics a year with carpal tunnel syndrome (G56.0 of Korean Standard Classification of Diseases), according to the statistics of the Korea Health Insurance Review and Assessment Service [33]. In addition, it is possible to observe various structures in the volar wrist, including nerves, arteries, tendons, and retinacula. We aimed to teach basic ultrasound functions using the diagnostic criteria for carpal tunnel syndrome, which includes the measurement of distance and cross-sectional area. Carpal tunnel syndrome is the most common entrapment syndrome of the upper extremity. Since nerve demyelination and fibrosclerosis may occur as it progresses, leading to sensory and motor deficits in the median nerve, early diagnosis is important. Medical professionals can quickly and accurately evaluate the compression of the median nerve in the carpal tunnel noninvasively using diagnostic ultrasound, and reliable diagnostic criteria have been established in several studies [25,34]. Therefore, this study was conducted because musculoskeletal ultrasound training is rare in traditional medicine education despite the high proportion of musculoskeletal disorders in traditional medicine clinical practice.

The results of this survey reinforce existing research findings that ultrasound can help identify anatomical structures by increasing the understanding of anatomy and physiology. As human body structures can be dynamically visualized using ultrasound, they are effectively used in anatomy and physiology education [5]. Since the safety of the procedure can be increased by using ultrasound, ultrasound has been used and reported to be effective in medical education for central venous catheter placement [35], peripheral intravenous techniques [36], and ultrasound-guided vascular access [37]. In addition, OSCE can assess learners’ progress and enhance learning in a clinical context [3]. For ultrasound OSCE, evaluation tools for abdominal ultrasound [38] and intensive care ultrasound [39] have been developed. Ultrasound training is necessary because the safety and effectiveness can be improved by checking the anatomical structures through the ultrasound guide and performing the procedure for the acupuncture, pharmacopuncture, and acupotomy used in traditional medicine [14,40,41,42]. Therefore, training and assessing students’ performance through the OSCE is essential in traditional medicine education for educational purposes. It is not limited to the procedural skills of ultrasound, but can also be used for anatomy, physiology, diagnostics, identifying acupoints, and performing acupuncture.

The importance of communicating with students about education and OSCE is significant and enables teachers to understand the needs of the learners [3]. In this study, most students showed high satisfaction with the education, although some students pointed out that the education time was short and that additional education was needed. The results of the majority of students stating that they are willing to study ultrasound more in the future are similar to those of previous studies on ultrasound education [37]. In this survey, the students asked for more ultrasound education in the shoulder, abdomen, knee, and ankle. According to the 2021 Korean medical institution frequent disease statistics of the Korea Health Insurance Review and Assessment Service, for the frequency of diseases of patients visiting traditional Korean medical institutions, dorsalgia ranked first, followed by dislocation, sprains, and strains of the lumbar spine and pelvis, neck, ankle and foot, shoulder girdle, and shoulder lesions ranked in the top 10 [33]. Therefore, there is a strong need for education on ultrasound-guided spine pharmacopuncture injection and shoulder diseases to increase the safety and accuracy of intervention [43]. In addition, one respondent requested guide needling for acupotomy. Since it is more invasive than conventional acupuncture, its side effects may be higher; therefore, education on ultrasound-guided acupotomy is recommended [44].

In this study, we unintentionally performed a non-randomized controlled study according to education time. In this ultrasound training, the learners’ training period was divided into 3 weeks and 8 weeks due to the spread of COVID-19. Although online lectures corresponding to steps 1 to 3 of George’s method [29] were provided to the 3-week group to adjust the educational conditions of the two groups as closely as possible, the total OSCE score was significantly higher in the 8-week group. There was no significant difference between the two groups in self-evaluation items, but in the survey, the students of the 8-week group answered that they had a sufficient OSCE time limit compared to the 3-week group. The difference between the two groups was that in the 8-week group, the ultrasound practice time (80 min) was longer than that in the 3-week group (60 min). In addition, the 8-week group experienced a mock OSCE with colleagues and submitted their scanned images to receive feedback from the preceptors, while the 3-week group did not. This can be understood in the same context as in the previous study, in which performance in the OSCE improved as the number of ultrasound scans increased [45]. It can be inferred that the longer the ultrasound experience and practice time, the better the performance of the OSCE.

### 4.3. Strengths and Limitations

The advantage of this study is that the educational effect was assessed subjectively and objectively after ultrasound training and OSCE for students majoring in traditional medicine, and it described specific ultrasound OSCE focusing on scanning the volar wrist and diagnosing carpal tunnel syndrome, which can be used in medical education. Most students had never used an ultrasound diagnostic device before, and they recorded high satisfaction in the post-education survey. The OSCE checklist and questionnaire used in this study were validated in our previous study [21]. The survey response rate was relatively high. In addition, since there was a group that received a 3-week face-to-face training including flipped learning and 8-week face-to-face training that did not include flipped learning, it was possible to compare the objective learning outcomes of the two groups.

The limitations of this study are as follows. First, because the education and survey were conducted by one college, the possibility of generalization is low. Second, practical training was only conducted for one disease, carpal tunnel syndrome, which may differ from the characteristics of other diseases. Third, in this ultrasound training, there were no pathological cases, because the students were asked to examine each other instead of using actual patients. Therefore, there is a limit to the lack of closeness when solving real-world problems. In addition, due to the COVID-19 pandemic, not all students received the same period of education, and the curriculum was adjusted. The response rate recorded in this survey was limited to 50%. It is considered that the response rate of 50% is not too low due to the nature of the survey study, and a plan to increase the response rate will be devised in future research.

### 4.4. Implications and Contributions for Further Education and Research

Based on the survey results, we found that ultrasound education is an essential part of traditional medicine student education. In the future, through a systematic ultrasound education curriculum, traditional medicine students should be educated to the level of ultrasound use for actual patients and reference them for treatment. Additional ultrasound training on other anatomical areas such as the shoulder, knee, and ankle is needed. In addition, a standardized teaching methodology is required, and ultrasound education should be prioritized to suit the clinical field of traditional medicine. Further research is needed to develop training content for ultrasound-guided interventions for musculoskeletal disorders and a specific OSCE checklist. In addition, it is necessary to conduct qualitative research on ultrasound education to analyze students’ needs in depth. Studies that target professors and students from various institutions are required.

## 5. Conclusions

The direct experience in the field improved the skills of the future Korean medical doctors and helped them understand the strengths and the limits of the ultrasounds. The OSCE score was higher in the group that practiced for 8 weeks than in the group that practiced for 3 weeks with flipped learning. Although a scientific comparative study on the learning curve was not planned, it was found that a sufficient training time was necessary considering the learning curve through an unintentional comparison of the number of training sessions. Further ultrasound training on other anatomical areas and guided interventions are needed for wider experience.

## Figures and Tables

**Figure 1 diagnostics-12-01707-f001:**
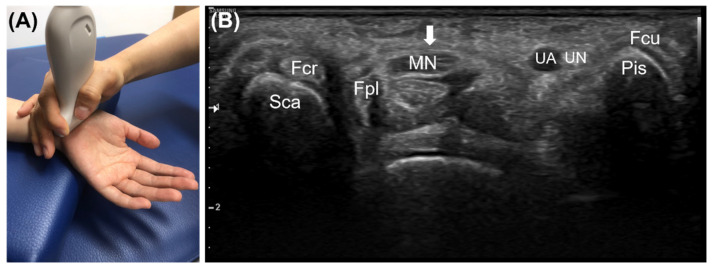
(**A**) Scanning the anterior aspect of the wrist. The ultrasound indicator is in the radial side. (**B**) Structures that can be observed with ultrasound in the proximal carpal tunnel: Transverse carpal ligament (arrow); Sca, scaphoid; Pis, pisiform; MN, median nerve; UN, ulnar nerve; UA, ulnar artery; Fcr, flexor carpi radialis; Fpl, flexor pollicis longus; Fcu, flexor carpi ulnaris. Tendons of the flexor digitorum superficialis and flexor digitorum profundus are located deep within the carpal tunnel.

**Figure 2 diagnostics-12-01707-f002:**
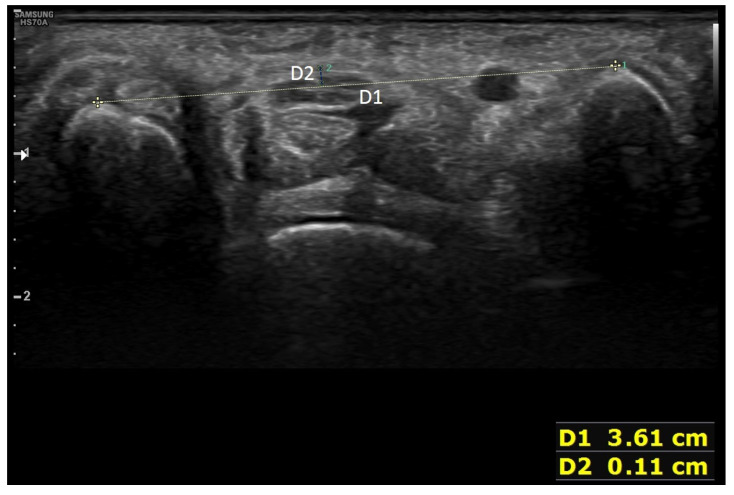
Diagnostic criteria for carpal tunnel syndrome by ultrasound relevant to palmar bowing of the flexor retinaculum. The yellow dot line of D1 shows a line connecting the top of the pisiform and the scaphoid bone. D2 is the distance from D1 to the transverse carpal ligament.

**Figure 3 diagnostics-12-01707-f003:**
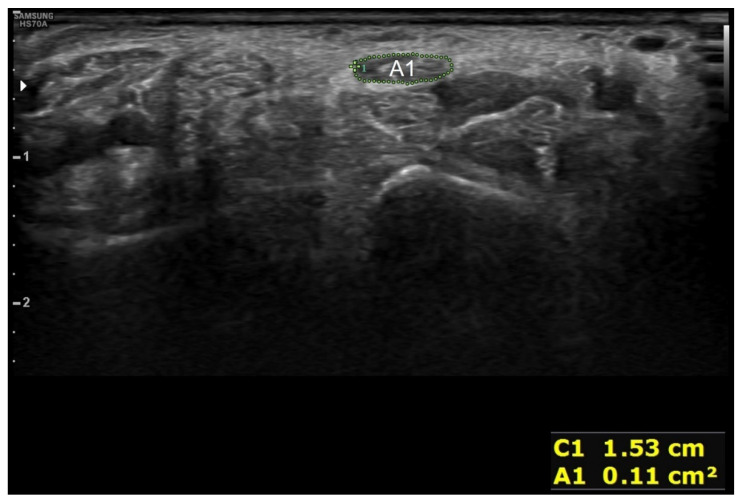
Diagnostic criteria for carpal tunnel syndrome by ultrasound relevant to the enlargement of the nerve proximal to the flexor retinaculum. A1 is the cross-sectional size of the median nerve, proximal to the flexor retinaculum.

**Table 1 diagnostics-12-01707-t001:** Comparison of opinions on ultrasound training and OSCE in the 3-week group and 8-week group.

Question (Answers Ranging from 1 (Not at All) to 5 (Entirely)	Total (*n* = 60)	3-Week Group (*n* = 37)		8-Week Group (*n* = 23)		*p*-Value
	Mean ± SD	Mean ± SD	Median (IQR)	Mean ± SD	Median (IQR)	
Were the learning objectives of the ultrasound training appropriate?	4.52 ± 0.57	4.49 ± 0.56	5 (4–5)	4.57 ± 0.59	5 (4–5)	0.530
Were the learning objectives and ultrasound training contents related well?	4.52 ± 0.57	4.49 ± 0.56	5 (4–5)	4.57 ± 0.59	5 (4–5)	0.530
Was it easy to understand the contents of the ultrasound training?	4.6 ± 0.49	4.59 ± 0.50	5 (4–5)	4.61 ± 0.50	5 (4–5)	0.914
Has ultrasound training improved your knowledge about the anatomy and physiology of carpal tunnels?	4.75 ± 0.44	4.68 ± 0.47	5 (4–5)	4.87 ± 0.34	5 (5–5)	0.094
Did the ultrasound practice help you develop ultrasound skills?	4.5 ± 0.62	4.43 ± 0.60	4 (4–5)	4.61 ± 0.66	5 (4–5)	0.182
Did the ultrasound OSCE help motivate independent practical training?	4.45 ± 0.53	4.41 ± 0.55	4 (4–5)	4.52 ± 0.51	5 (4–5)	0.448
Was the ultrasound OSCE difficult?	2.98 ± 1.11	2.92 ± 1.06	3 (2–4)	3.09 ± 1.20	3 (2–4)	0.569
Was the ultrasound OSCE time limit (5 min) enough for you?	3.27 ± 1.22	2.97 ± 1.30	3 (2–4)	3.74 ± 0.92	4 (3–4)	0.019 *
Are you generally satisfied with this ultrasound training?	4.5 ± 0.60	4.41 ± 0.60	4 (4–5)	4.65 ± 0.57	5 (4–5)	0.091
Do you have any intentions of studying ultrasound more in the future?	4.57 ± 0.56	4.57 ± 0.60	5 (4–5)	4.57 ± 0.51	5 (4–5)	0.804
Are you willing to use ultrasound in clinical practice after graduation?	4.10 ± 0.71	4.08 ± 0.76	4 (4–5)	4.13 ± 0.63	4 (4–5)	0.912
Has your opinion about ultrasound usage changed positively through this education?	4.33 ± 0.60	4.30 ± 0.66	4 (4–5)	4.39 ± 0.50	4 (4–5)	0.706

SD, standard deviation. Mann–Whitney U test was performed. * *p* < 0.05.

**Table 2 diagnostics-12-01707-t002:** Comparison of self-assessment of ultrasound competency and attitude in the 3-week group and 8-week group.

Items of Self-Assessment (Answers Ranging from 1 = Strongly Disagree to 5 = Completely Agree)	Total (*n* = 60)	3-Week Group (*n* = 37)		8-Week Group (*n* = 23)		*p*-Value
	Mean ± SD	Mean ± SD	Median (IQR)	Mean ± SD	Median (IQR)	
I actively participated in ultrasound training.	4.40 ± 0.74	4.27 ± 0.80	4 (4–5)	4.61 ± 0.58	5 (4–5)	0.089
I was considerate of the subject during the ultrasound examination.	4.35 ± 0.66	4.30 ± 0.70	4 (4–5)	4.43 ± 0.59	4 (4–5)	0.511
I communicated appropriately with the subject during the ultrasound examination.	4.45 ± 0.59	4.38 ± 0.64	4 (4–5)	4.57 ± 0.51	5 (4–5)	0.303
I can use basic functions of ultrasound such as ultrasonic probe selection, depth and focus control, and freeze.	4.52 ± 0.70	4.41 ± 0.80	5 (4–5)	4.70 ± 0.47	5 (4–5)	0.203
I can identify the anterior structure of the wrist through ultrasound.	4.35 ± 0.71	4.24 ± 0.80	4 (4–5)	4.52 ± 0.51	5 (4–5)	0.232
I can do ultrasound scans according to the criteria for diagnosing carpal tunnel syndrome.	4.40 ± 0.67	4.30 ± 0.74	4 (4–5)	4.57 ± 0.51	5 (4–5)	0.192

SD, standard deviation; IQR, interquartile range. Mann–Whitney U test was performed.

**Table 3 diagnostics-12-01707-t003:** OSCE checklist for the ultrasound scan of the volar wrist and diagnosis of carpal tunnel syndrome. Students’ OSCE scores also presented as mean and standard deviation.

Task	Score	Sub-Score	Mean ± SD
1. Self-introduction	1		0.99 ± 0.10
2. Patient identification	1		0.96 ± 0.19
3. Explaining the ultrasound examination to the patient	1		0.98 ± 0.13
4. Hand hygiene before the examination	2		1.46 ± 0.59
5. Patient guidance	2		1.86 ± 0.34
6. Selecting the appropriate probe and setting the device	3		1.43 ± 1.28
6–1. Probe setting		1	0.57 ± 0.50
6–2. Selecting the area to be scanned		1	0.57 ± 0.50
6–3. Adjusting the depth to scan		1	0.28 ± 0.45
7. Applying the gel	1		1.00 ± 0.00
8. Probe tuning	1		0.91 ± 0.29
9. Probe handling	1		0.83 ± 0.38
10. Correct view of the (1) scaphoid, (2) pisiform, (3) median nerve, (4) ulnar artery, (5) flexor retinaculum, (6) flexor pollicis longus, (7) ulnar nerve, (8) flexor digitorum superficialis, flexor digitorum profundus, (9) flexor carpi radialis, (10) flexor carpi ulnaris	3		2.33 ± 0.74
10–1. 9–10 structures		3	
10–2. 4–8 structures		2	
10–3. 1–3 structures		1	
10–4. 0 structure		0	
11. Correct measurement according to diagnostic criteria of carpal tunnel syndrome and explaining the results	3		2.21 ± 0.97
11–1. Connecting the top of the scaphoid and the pisiform with a line		1	0.76 ± 0.43
11–2. Measuring the vertical distance between the line connecting the top of the scaphoid and the pisiform and the flexor retinaculum		1	0.64 ± 0.48
11–3. Explaining the examination results		1	0.81 ± 0.39
12. Correct measurement of the cross-sectional area of the median nerve and explaining the examination results	3		1.65 ± 1.19
12–1. Correct measurement of the cross-sectional area of the median nerve proximal to the flexor retinaculum		2	1.07 ± 0.83
12–2. Explaining the examination results		1	0.58 ± 0.50
13. Hand hygiene after the examination?	2		0.30 ± 0.66
14. Informing the patient about the examination’s completion and explaining the final findings after the examination	2		1.05 ± 0.96
15. Cleaning up the used devices and wiping off the gel	2		1.04 ± 0.95
Total score	28		19 ± 5.35

OSCE, Objective Structured Clinical Examination; SD, standard deviation.

## Data Availability

The data presented in this study are available upon reasonable request.

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
