# Peer review of "Implementation of Objective Structured Clinical Examination on Diagnostic Musculoskeletal Ultrasonography Training in Undergraduate Traditional Korean Medicine Education: An Action Research"

_diagnostics, 2022, doi:10.3390/diagnostics12071707_

Round 1

Reviewer 1 Report

OK, I think the learming has to be greater when it is 8 weeks versus 3. Now, maybe 8 weeks is too long versus 3.

Author Response

[# Response to Reviewer 1 Comments]

  • Comment 1-1

OK, I think the learming has to be greater when it is 8 weeks versus 3. Now, maybe 8 weeks is too long versus 3.

# Response 1-1

Thank you for your comment. We agree with the comment that 8 weeks is too long versus 3.

This ultrasound training was provided once a week for 8 weeks. This was described in section 2.1 (page 2). Therefore, it is actually 8 times versus 3 times. Due to the COVID-19 situation, the training period was unintentionally changed, from 8 weeks to 3 weeks for two groups out of three.

In our opinion, 8-times training was relatively adequate for this ultrasound education contents. 3-times training was too short on time, so we provided relevant knowledge by online lectures in advance so that the learner could focus on the practice.

In the future, we plan to train students under the same conditions. Concerning future improvement, the Conclusions section has been revised as follows.

(See also page 11)

The direct experience on the field improved the skill of the future Korean medicine doctors and helped them understand the strengths and the limits of the ultrasounds. The OSCE score was higher in the group that practiced for 8 weeks than in the group that practiced for 3 weeks with flipped learning. Although a scientific comparative study on the learning curve was not planned, it was found that sufficient training time was necessary considering the learning curve through an unintentional comparison of the number of training sessions. Further ultrasound training on other anatomical areas and guided interventions are needed for wider experience.

Reviewer 2 Report

The study "Implementation of Objective Structured Clinical Examination on Diagnostic Musculoskeletal Ultrasonography Training in Undergraduate Traditional Korean Medicine Education: An Action Research" is not a classical scientific paper on the ultrasound application in oncological or orthopaedic practice, but is an investigation on  two different methods to improve the " learning curve"  of the students on ultrasound practice for pathology of the wrist.

The  results are as expected: the agreement was high, students with longer training ( 8 weeks) did better  than the students with a shorter practical course ( 3 weeks).

The direct experience on the field improved the skilling of the future doctors to understand the strenght and the limits of the ultrasounds.

A larger number of young attendants ( in this study were  involved only 111 students)   and a higher number of responses to the questionnaire are recommended .

A wider experience  on other anatomical areas is  advisable in order to increase the confidence of the students on ultrasound exams.

Author Response

[# Response to Reviewer 2 Comments]

  • General Comment 2-0

The study "Implementation of Objective Structured Clinical Examination on Diagnostic Musculoskeletal Ultrasonography Training in Undergraduate Traditional Korean Medicine Education: An Action Research" is not a classical scientific paper on the ultrasound application in oncological or orthopaedic practice, but is an investigation on  two different methods to improve the " learning curve"  of the students on ultrasound practice for pathology of the wrist.

The  results are as expected: the agreement was high, students with longer training ( 8 weeks) did better  than the students with a shorter practical course ( 3 weeks).

The direct experience on the field improved the skilling of the future doctors to understand the strenght and the limits of the ultrasounds.

# Response 2-0

Thank you for your careful review on our manuscript. The following sentence was added in the Conclusion.

(See also page 11)

Although a scientific comparative study on the learning curve was not planned, it was found that sufficient training time was necessary considering the learning curve through an unintentional comparison of the number of training sessions.

  • Comment 2-1

A larger number of young attendants ( in this study were  involved only 111 students)   and a higher number of responses to the questionnaire are recommended .

# Response 2-1

In our department, there are about 100 students per grade, and the training was provided for one grade of students. Referring to this OSCE experience, we plan to expand ultrasound education to other anatomical areas for several grades in the future. We have added the need for higher number of responses in the Discussion (Strengths and Limitations) section.

(See also page 10-11)

The response rate recorded in this survey was limited to 50%. It is considered that the response rate of 50% is not too low due to the nature of the survey study, and a plan to increase the response rate will be devised in future research.

  • Comment 2-2

A wider experience  on other anatomical areas is  advisable in order to increase the confidence of the students on ultrasound exams.

# Response 2-2

Thank you for your comment. We totally agree that this ultrasound training should be extended to more students and cover other anatomical areas for wider experience. Our conclusion has been revised following your opinion.

(See page 11)

Additional ultrasound training on other anatomical areas such as shoulder, knee, ankle is needed.

(See also page 11)

Further ultrasound training on other anatomical areas and guided interventions are needed for wider experience.

Reviewer 3 Report

Dear authors,

Please find the following comments:

- Please re-write the conclusion to support  your obtained results.

- Please add "ethical approval" statement.

- Along the manuscript, English editing is required.

Best Regards.

Author Response

[# Response to Reviewer 3 Comments]

  • Comment 3-1

Dear authors,

Please find the following comments:

- Please re-write the conclusion to support your obtained results.

# Response 3-1

Thank you for your comment. The conclusion has been revised based on the findings of our study. The following is our revised conclusion.

(See also page 11)

The direct experience on the field improved the skill of the future Korean medicine doctors and helped them understand the strengths and the limits of the ultrasounds. The OSCE score was higher in the group that practiced for 8 weeks than in the group that practiced for 3 weeks with flipped learning. Although a scientific comparative study on the learning curve was not planned, it was found that sufficient training time was necessary considering the learning curve through an unintentional comparison of the number of training sessions. Further ultrasound training on other anatomical areas and guided interventions are needed for wider experience.

  • Comment 3-2

- Please add "ethical approval" statement.

# Response 3-2

Thank you for your comment. The ethical approval statement was added in the Methods section (Study design and participants).

(See also page 5)

This study was approved by the Institutional Review Board of Wonkwang University (WKIRB-202109-SB-069).

  • Comment 3-3

- Along the manuscript, English editing is required.

Best Regards.

# Response 3-3

Thank you for your comment. This manuscript had undergone English language editing by Editage (www.editage.co.kr). Please see the attachment (editing certificate).

Thank you.
